# Alkaloid Concentrations of *Lolium perenne* Infected with *Epichloë festucae* var. *lolii* with Different Detection Methods—A Re-Evaluation of Intoxication Risk in Germany?

**DOI:** 10.3390/jof6030177

**Published:** 2020-09-18

**Authors:** Veronika Vikuk, Benjamin Fuchs, Markus Krischke, Martin J. Mueller, Selina Rueb, Jochen Krauss

**Affiliations:** 1Department of Animal Ecology and Tropical Biology, University of Würzburg, 97074 Würzburg, Germany; selina.rueb1@stud-mail.uni-wuerzburg.de (S.R.); j.krauss@uni-wuerzburg.de (J.K.); 2Biodiversity Unit, University of Turku, 20014 Turku, Finland; benjamin.fuchs@utu.fi; 3Department of Pharmaceutical Biology, Metabolomics Core Unit, University of Würzburg, 97074 Würzburg, Germany; krischke@biozentrum.uni-wuerzburg.de (M.K.); martin.mueller@biozentrum.uni-wuerzburg.de (M.J.M.)

**Keywords:** *Epichloë*, *Lolium perenne*, toxicity, grasslands, HPLC/UPLC methods, endophyte, plant fresh/dry weight, alkaloid detection methods, mycotoxins, phenology

## Abstract

Mycotoxins in agriculturally used plants can cause intoxication in animals and can lead to severe financial losses for farmers. The endophytic fungus *Epichloë festucae* var. *lolii* living symbiotically within the cool season grass species *Lolium perenne* can produce vertebrate and invertebrate toxic alkaloids. Hence, an exact quantitation of alkaloid concentrations is essential to determine intoxication risk for animals. Many studies use different methods to detect alkaloid concentrations, which complicates the comparability. In this study, we showed that alkaloid concentrations of individual plants exceeded toxicity thresholds on real world grasslands in Germany, but not on the population level. Alkaloid concentrations on five German grasslands with high alkaloid levels peaked in summer but were also below toxicity thresholds on population level. Furthermore, we showed that alkaloid concentrations follow the same seasonal trend, regardless of whether plant fresh or dry weight was used, in the field and in a common garden study. However, alkaloid concentrations were around three times higher when detected with dry weight. Finally, we showed that alkaloid concentrations can additionally be biased to different alkaloid detection methods. We highlight that toxicity risks should be analyzed using plant dry weight, but concentration trends of fresh weight are reliable.

## 1. Introduction

Mycotoxins are secondary metabolites produced by fungi, which can cause disease and death in humans and other vertebrates, but also in invertebrates, plants, and microorganisms [1,2]. Fungal endophytes of the genus *Epichloë* live symbiotically and asymptomatically inside cool season grass species [3]. They can produce different alkaloids, which can be toxic for vertebrates or invertebrates and provide protection from herbivores for the plant [4,5]. The plant provides shelter, nutrition, and dispersal for the fungus [5], whereas the fungus increases plant fitness, biomass, and drought resistance [6,7].

One economically important grass endophyte is *Epichloë festucae* var. *lolii* infecting *Lolium perenne* L. (perennial ryegrass) [8]. The systemic fungus reproduces exclusively asexual via the seed of its host plant [4,5]. The fungus–grass symbiosis is able to produce the pyrrolopyrazine peramine, an insect-deterring alkaloid, as well as the vertebrate toxic alkaloids lolitrem B, an indole-diterpene, its precursor paxilline, and the ergot alkaloid ergovaline [8,9,10,11]. Biologically active concentrations for peramine vary between 2 µg/g dry weight (DW), when Argentine stem weevil larvae fed on artificial diets [12,13] and 15 µg/g in planta for a strong resistance against Argentine stem weevil [14,15]. Lolitrem B and ergovaline showed signs of intoxication in livestock from 1.8 µg/g (DW) and 0.3 µg/g (DW), respectively, based on plant dry weight [16,17]. Lolitrem B causes intoxications of livestock, known as ryegrass staggers [8,10,18]. Severe outbreaks of ryegrass staggers are known from New Zealand and Australia [8]. In Australia, approximately 100,000 animals died in 2002 due to ryegrass staggers caused by lolitrem B produced by *E. festucae* var. *lolii* infecting *L. perenne* (Hume et al., 2016). Financial losses for farmers caused by *Epichloë* endophyte intoxication events are estimated worldwide at around 2 billion US Dollar per year (Hume et al., 2016). Toxic secondary plant compounds can be also a thread in the human food chain, such as pyrrolyzidin alkaloids in honey [19]. However, studies showed that concentrations of *Epichloë* alkaloids in the milk of intoxicated cows or in sheep fat are below the toxicity threshold, indicating that the intoxication risk of humans consuming animal products such as milk or meat is low [20,21]. Perennial ryegrass (*L. perenne*) is a commonly used forage grass species due to its fast initial growth and persistence [22]. Hence, an evaluation of intoxication risks caused by *Epichloë* infections is important for farmers.

In Europe, intoxication events caused by *Epichloë* endophytes are scarce and intoxication risk in Germany is considered to be low [23,24], explained by low infection rates (15%) [24] compared to New Zealand (70%) [25] and more heterogenous grasslands [24,26]. Abiotic factors such as drought and elevated temperature can influence infection rates of grasses with *Epichloë* endophytes and alkaloid concentrations [6,27] as well as land-use intensity [28,29]. However, a previous study conducted on German grasslands in 2015 showed no influence of land-use intensity on infection rates or alkaloid concentrations [23].

The main method to estimate intoxication risks is the quantitation of alkaloids in grasses with analytical chemistry methods. However, methodologies differ substantially between studies. Most studies used high-performance liquid chromatography (HPLC) with different detection methods like mass spectrometry [30], tandem mass spectrometry [24,31,32], or fluorescence detection [16,20,21,33,34] for the quantitation of lolitrem B, paxilline, ergovaline, and peramine, while one study used enzyme immunoassays [35]. Analytical reference substances are used for the validation of analytical method or as external standard for quantitation of alkaloids (e.g., lolitrem B in [20,21]) but are rarely commercially available. For quantitation of alkaloids as well, structurally similar substances (e.g., homoperamine for peramine [23,24,31,36] or stable isotope-labelled compounds are used as internal standard [32]. To quantify the alkaloids either fluorometric response curves [33] or fluorescence detectors [37] are further applied possibilities [37]. Homoperamine has been used as internal standard for quantitation of lolitrem B, although for a non-validated quantitation due to the structural differences [23,24,31,36], whereas ergovaline is often quantified with ergotamine as internal standard compound [23,24,31,36,38,39,40], as well as with lysergic acid diethylamide-d3 (LSD-d3) [32]. Standard calibration curves in combination with HPLC-fluorescence analyses can also be used for the quantitation of ergovaline [17,39]. In addition, the original plant material sampled and processed for the extraction is important. Few studies used plant fresh weight to analyze alkaloid concentrations [23,36], whereas the use of plant dry weight is the standard to detect toxicity of compounds [16,34,38]. Sometimes plant fresh weight was used for analytical detection, if the target substance degrades or chemically changes (e.g., some phytohormones) when drying the plant samples in the oven [41,42] or if the amount of plant material is limited [23]. However, using plant fresh weight complicates comparison of alkaloid concentrations between different studies. Hence, plant samples are often freeze-dried, because this method protects susceptible samples and solves the problems of varying amount of water. However, there are also studies that used straw for alkaloid analyses [16,34]. Many of the published toxicity thresholds for peramine, lolitrem B, and ergovaline are based on analyses of plant dry weight. In best practice, new methods need to be validated, compared to other licensed methods, and tested for their robustness [32,43], before they can be licensed for the quantitation of alkaloids. However, there can be essential differences in methods, which might lead to different concentrations. Bauer et al. (2018) showed for example that paxilline concentrations detected with enzyme immunoassays differed compared to measurements with HPLC-MS/MS analysis.

In this study, we follow the research of König et al. (2018) now using dry weight instead of fresh weight to (i) confirm or detect changes of infection rates and alkaloid concentrations in German grasslands. Additionally, we (ii) analyzed seasonal changes in alkaloid concentrations of *L. perenne* on five selected real-world grasslands in northern Germany for ten months during the growing period. Finally, we conducted a common garden experiment with a commercially available cultivar of *L. perenne* infected with *E. festucae* var. *lolii*, to (iii) detect seasonal changes in alkaloid concentrations and to (iv) compare changes in alkaloid concentrations detected with plant fresh and dry weight over six months. Besides using plant fresh and dry weight, we applied two different UPLC (Ultra Performance Liquid Chromatography) methods [31,36] to (v) show changes in alkaloid concentrations depending on the detection method.

## 2. Materials and Methods

### 2.1. Dataset 1: Field Study in Germany

Our field study covered 150 study sites in three regions, which are part of a nation-wide long-term study in Germany, called Biodiversity Exploratories (www.biodiversity-exploratories.de). We conducted our field study in three regions: UNESCO biosphere region Schwäbische Alb (South-West Germany), national park Hainich (Central Germany), and UNESCO biosphere region Schorfheide (North-West Germany) [24]. The sampled regions cover different landscape types and land-use intensity gradients, further spanning a latitudinal area of 800 km from north to south within Germany [44]. Each study site is 50 m × 50 m in size within grassland areas [44]. Land-use intensity (LUI) was calculated with a formula, developed for the Biodiversity Exploratories integrating fertilization, mowing, and grazing [45]. The influence of LUI was tested on infection rates of *L. perenne* with *Epichloë festucae* var. *lolii* sampled in 2017 and the produced alkaloids. Since fertilization, mowing, and grazing can also have separate influence on infection rates and alkaloid concentrations [23,46]; we also analyzed them separately. We monitored all study sites from May until August 2017 for the occurrence of *Lolium perenne* and sampled 20 individual plants per study site, where enough individuals were found. We sampled individuals with a minimum distance of three meters to avoid sampling the same individual twice. In 2017, we sampled the same study sites as König et al. (2018) in 2015 to compare the effects of land-use intensity and region on infection rates and alkaloid concentrations. One tiller per plant individual was sampled in the previous survey in 2015 [23]; three were sampled in 2017. Each sample was taken approximately 3 cm from the lowest part of the grass tiller, because alkaloids and *Epichloë* endophyte accumulate there [23,47]. We collected the samples in 2 mL Eppendorf tubes, kept them on dried ice in the field, and stored them at −20 °C before further processing. Infection rates of *L. perenne* with *Epichloë festucae* var. *lolii* were detected with specific primers by a polymerase chain reaction (PCR) method in all collected samples from 2017 as described in [24]. After PCR, only positive samples were tested for the presence of alkaloids. Alkaloid concentrations were calculated on population level (infected and not infected samples per study site) and individual level.

### 2.2. Dataset 2: Alkaloid Concentrations throughout the Year in a Field Study

We sampled *L. perenne* plants on five different study sites in the region Schorfheide-Chorin, northern Germany in 2018, which were previously confirmed with high alkaloid concentrations of *Epichloë* infected *L. perenne* individuals in 2015 [23] (Appendix A) in order to detect seasonal changes in the alkaloid production. We sampled three tillers of 10–20 plants per study site every two months, at the beginning of April and June and at the end of July, September, November 2018, and January 2019. Infection rates were checked with a commercially available kit for immunoblot assays (www.agrinostics.com) and only infected grass samples were tested for alkaloids. Alkaloid concentrations, quantified with dry weight, were calculated on population level (mean alkaloid concentration of infected and not infected samples per study site) and for infected samples only. We calculated means and standard errors of the five study sites to show seasonal trends.

### 2.3. Dataset 3: Common Garden Study

In 2019, we conducted a common garden experiment with sown *Epichloë* infected *L. perenne* plants to detect seasonal changes in alkaloid production in six months during the main growing season in central Europe (April until September). We used the same experimental setup as described in Fuchs et al. (2017b) with some minor changes: On 9 January 2019, we planted 216 endophyte-infected *Lolium perenne* seeds (A26558, cv. Grasslands Samson ‘common toxic’ [48]) in 4 trays, each tray with 54 pots (diameter 5 cm, height 5 cm) and one tray with 54 endophyte-free seeds of the same plant cultivar (A11104, Grassland Samson Nil endo). Seeds were provided by AgResearch, New Zealand. The substrate was a potting compost (Plantiflor Pro Natur Bio Quality), and pots were positioned in a greenhouse (day: 5 a.m.–9 p.m., temperature: 20 °C, RH: 50%; night: 9 p.m.–5 a.m., temperature: 15 °C, RH: 75%) at the University of Würzburg, Germany. After six weeks of growth, plants were repotted in common garden soil (Einheitserde classic CLED73, Profi Substrat) in single pots (11 × 11 × 11 cm). After 14 weeks in the greenhouse, the plants were again repotted in bigger pots (17 × 17 × 17 cm) and placed randomly in a common garden setup at the University of Würzburg, Germany for the entire experimental period.

We sampled biweekly ten random endophyte-infected plants from 9 April to 25 September 2019. Sampling was conducted by cutting randomly 10 complete grass tillers from each pot, which were immediately frozen and kept at −20 °C until further preparation for chemical analyses. For alkaloid detection the plant material of each sample was ground and freeze-dried for 72 h. Sampled plants were marked to avoid sampling the same plant twice. The first plant samples were taken after 13 weeks of growth in the greenhouse, before the plants were located in the common garden, in order to be able to compare alkaloid concentrations in the greenhouse with the common garden. For ten sampling points (April–September), samples were divided after homogenization and half of the sample was analyzed with plant fresh weight, whereas the other half was freeze-dried. Therefore, we were able to directly compare alkaloid concentrations in fresh and dried plant weight. Alkaloid detections were performed with approximately 40 mg plant fresh weight and approximately 20 mg freeze-dried plant weight.

### 2.4. Alkaloid Detection Methods

Alkaloid concentrations for datasets (1) and (2), as well as for the comparison of fresh and dry weight concentrations in dataset (3), were detected by using the following method (method 1) (Appendix A): Quantitation of peramine, lolitrem B, ergovaline, and paxilline was performed by ultra-high-performance liquid chromatography coupled with tandem mass spectrometry (UPLC-MS/MS) using a Waters Acquity UPLC combined with a Quattro Premier triple quadrupole mass spectrometer. After grinding and freeze drying for 72 h, grass samples were prepared as described in a previous publication [31]. We used approximately 20 mg of freeze-dried plant material. After extraction with 250 µL dichloromethane/methanol 1:1 (*v*/*v*) containing the internal standards ergotamine (500 ng), lysergic acid diethylamide-d3 (LSD-d3) (125 ng), and homoperamine (500 ng) and a centrifugation (14,000 rpm, 10 min), the supernatant was recovered, and the pellet was reextracted with 250 μL of dichlormethane/methanol 1:1 (*v*/*v*). The organic phases were combined and two aliquots of the supernatants were evaporated under reduced pressure. One sample was reconstituted in water/acetonitrile/formic acid 50:50:0.1 (*v*/*v*/*v*) for the analysis of lolitrem B and paxilline, and the other in water/acetonitrile/formic acid 80:20:0.1 (*v*/*v*/*v*) for the analysis of peramine and ergovaline. The solubility and the chromatographic separation of lolitrem B and paxilline was better in water/acetonitrile/formic acid 50:50:0.1 (*v*/*v*/*v*), whereas the chromatographic separation of peramine and ergovaline needed a higher water content (water/acetonitrile/formic acid 80:20:0.1 (*v*/*v*/*v*)).

Chromatographic separation was conducted according to a published protocol with slight modifications [32]. An Acquity UPLC column (100 × 2.1 mm; 1.7 µm; Waters GmbH, Eschborn, Germany) was used. An amount of 0.1% formic acid dissolved in water (solvent A) and acetonitrile containing 0.1% formic acid (solvent B) was used for the following gradient elution: 0–1.0 min 98% solvent A, 1.0–3.0 min to 90% solvent A, 3.0–5.0 min to 85% solvent A, 5.0–7.5 min to 80% solvent A, 7.5–10.0 min to 75% solvent A, 10.0–11.5 min to 70% solvent A, 11.5–13.0 min to 65% solvent A, 13.0–14.5 min to 50% solvent A, 14.5–16.0 min to 40% solvent A, 16.0–19.0 min to 0% solvent A, 19.0–22.0 hold 0% solvent A, 22.0–23.0 back to 98% solvent A, and hold for another 2 min (total run time 25 min). Injection volume was 10 µL for the analyses of peramine and ergovaline and 5 µL for the analyses of lolitrem B and paxilline. Parameters for alkaloid detection were set as in Fuchs et al. (2013), using multiple reaction monitoring (MRM). Additionally, LSD-d3 was used as internal standard with the following specific transitions: m/z 327.1→208.1 and 327.1→226.1. Paxilline was detected using the following transitions: m/z 436.3→130.1 and 436.3→182.1. Following the protocol of [32], we tested LSD-d3 as internal standard for the quantitation of ergovaline, but quantitation with ergotamine as internal standard gave more reliable results due to a better similarity concerning the physicochemical properties of both compounds (retention time, ionization efficiency, and fragmentation of ergotamine were more similar to ergovaline). In method 1, the limit of quantitation (LOQ) for paxilline and lolitrem B was 0.5 ng on column and 0.1 ng on column for ergovaline and peramine, respectively.

In dataset (3), we additionally compared alkaloid concentrations by using dry plant material determined with method 1, as described in this paper and an UPLC-MS/MS method based on parameters published in [36] (method 2) (Appendix A). Method 1 was optimized to detect a higher number of endophyte-derived alkaloids similar to lipophilic alkaloids, such as paxilline and other *Epichloë* alkaloids, like lolines, in one analysis. We used dried plant material from week 29 (cut number 8) and detected the same ten samples with both methods in order to evaluate if methods produce comparable results and can be used interchangeably. Methodological differences in method 2 were as following: Samples were diluted in 80% methanol before HPLC measurements to allow analysis of lolitrem B, peramine, and ergovaline from one sample. Data for the detection of paxilline were not acquired within this method. Another column (Acquity BEH column (50 × 2.1 mm; 1.7 µm; Waters GmbH, Eschborn, Germany)) was used for separation at a flow rate of 0.3 mL/min, with the same solvents (A + B) as method 1, but with a different gradient elution: from 5% to 25% solvent B in 5 min, followed by 25% to 75% solvent B in 0.5 min, then 75% to 100% solvent B in 2.5 min (total run time 10 min). Chromatograms of the analytes of both methods are shown in the Appendix A (Appendix A).

### 2.5. Statistical Analyses

Statistical analyses were conducted using the statistical software R (version 3.5.2). We tested the effects of land-use intensity (LUI) and region on infection rates, followed by additional models replacing LUI by either mowing, grazing, or fertilization using a generalized linear model (glmer, package: lme4) [49] with study site as a random effect. The model was followed by a Wald chi-square test (ANOVA: type II, Wald Chi-square (χ^2^) tests, package: Car) [50]. Difference between regions were tested using a Tukey post-hoc test.

The response variables peramine and lolitrem B concentrations were analyzed with the same explanatory variables using linearized mixed effect models (lmer, package: lmerTest [51]) with the study site as random effect. We only tested samples statistically, where alkaloid concentrations were detected by UPLC-MS/MS. Samples that showed no alkaloids were excluded. The response variable alkaloid concentration was ln-transformed to improve homoscedasticity and normality of residuals. In the *L. perenne* samples from 2017, ergovaline was detected in only three samples; because of that, the variable ergovaline was not analyzed statistically.

Differences between alkaloid concentrations detected with two analytical methods (method 1 and method 2) were compared with a paired *t*-test.

### 2.6. Data Availability

The related raw data were deposited on the BExIS database of the Biodiversity Exploratories (www.bexis.uni-jena.de) with the following dataset IDs: 26046 and 26047.

## 3. Results

### 3.1. Dataset 1: Field Study in Germany

In 2017, we collected and analyzed 1122 *L. perenne* individuals on 66 grassland study sites in the three study regions. Neither infection rates nor alkaloid concentration were significantly affected by land-use intensity, mowing, grazing, or fertilization (Table 1). Significantly more infections occurred in the regions Schorfheide-Chorin (SCH) and Hainich (HAI) compared to the Schwäbische Alb (ALB), but alkaloid concentrations were not significantly different (*p* > 0.05) (Table 1).

In ALB, we detected concentrations of 3.38 ± 1.90 µg/g (DW) of peramine (mean ± SE) and 1.51 ± 1.17 µg/g (DW) of lolitrem B, but due to low samples size (*n* = 3), we did not analyze it statistically. In HAI, a mean peramine concentration of 2.23 ± 0.36 µg/g (DW) and a mean lolitrem B concentration of 0.61 ± 0.10 µg/g (DW) were detected. In SCH, mean concentrations of peramine were 1.25 ± 0.12 µg/g (DW) and for lolitrem B 0.34 ± 0.06 µg/g (DW). Although alkaloid concentrations were extracted from dry weight, alkaloid concentrations on population level were below toxicity thresholds for insect and vertebrates, similar to fresh weight concentrations in [23]. At an individual level, 26 *L. perenne* samples showed dry weight peramine concentrations above toxicity threshold for insects, and seven vertebrate-toxic lolitrem B concentrations exceeded the threshold (Table 2).

### 3.2. Dataset 2: Alkaloid Concentrations throughout the Year in a Field Study

In 2018, we sampled 575 *L. perenne* plants on five study sites in region SCH throughout the year. In 2015, these five study sites showed the highest alkaloid concentrations in a study with 87 study sites (Appendix A) [23].

133 out of 575 samples were immunoblot positive in 2018 and were analyzed for their alkaloid concentrations, as well as 12 not infected samples as negative controls.

Alkaloid concentrations could not be analyzed for 10 infected samples, due to limited amount of plant material and were excluded from the following analyses. None of the UPLC-MS/MS-analyzed immunoblot-negative samples showed detectable alkaloid concentrations. In eight immunoblot-positive samples no alkaloids were detected. In the remaining 116 infected samples we detected peramine concentrations ranging between 0.04 and 23.38 µg/g (DW). Peramine concentrations above the toxicity threshold of 2 µg/g (DW) were detected in 65 of the samples, thereby 27 were collected in September, 15 in July, 12 in June, 7 in November, and 5 in January. Mean peramine concentration per month on population level for September was with 3.23 ± 0.61 µg/g (DW) (mean ± SE) above the toxicity threshold (Figure 1, Appendix A), in three of the study sites SEG43 (5.11 ± 1.63 µg/g (DW)), SEG44 (2.66 ± 1.46 µg/g (DW)), and SEG46 (7.34 ± 1.43 µg/g (DW)). In 76 of the endophyte-infected samples, we detected lolitrem B with concentrations ranging between 0.07 and 23.81 µg/g (DW) of which 31 contained concentrations above the toxicity threshold of 1.8 µg/g (DW). All samples with concentrations above toxicity threshold for lolitrem B were sampled during the summer months (June, July, September). However, on population level all lolitrem B concentrations, per month (Figure 1) and per study site, were below toxicity threshold. Paxilline was not detected in the samples, and ergovaline was only detected in four samples and therefore not statistically analyzed. Ergovaline concentrations in all four samples were below the specific toxicity threshold of 0.3–0.4 µg/g (DW). Considering only infected samples (Figure 1, filled dots), peramine concentrations exceeded the toxicity threshold in July, September, and November, and lolitrem B concentrations exceeded the threshold in September. This result indicates that monocultures containing exclusively *Epichloë* infected individuals can be toxic to livestock in late summer.

### 3.3. Dataset 3: Common Garden Study

Alkaloid concentrations in *L. perenne* seeds infected with *E. festucae* var. *lolii* from New Zealand varied monthly during the study period (April–September) in our common garden experiment (Figure 2). In all samples (fresh and dry weight), concentrations of peramine exceeded the estimated toxicity threshold of 2 µg/g (DW), except in calendar week 21. The first samples were taken during plant rearing in the green house, before plants were placed to the common garden (dashed vertical line) starting with a mean concentration of 26.21 ± 2.97 µg/g dry weight (DW) (mean ± SE). Concentration decreased until calendar week 21 to 1.02 ± 0.10 µg/g (DW) peramine, increased again until calendar week 33 (August) to 35.63 ± 3.10 µg/g (DW), and decreased again to 16.32 ± 2.30 µg/g (DW) at the end of September (Figure 2, Appendix A). The concentrations analyzed from fresh weight showed a similar trend as plant dry weight but were only 24–37% of the concentrations determined in dry weight (averaged approx. 33% of dry weight).

Ergovaline concentration reached the peak in calendar week 25 (June) with 1.33 ± 0.30 µg/g (DW), and this level was maintained until end of September when concentrations decreased to 0.81 ± 0.18 µg/g (DW) (Figure 2, Appendix A). Concentrations analyzed based on plant dry weight were all above toxicity threshold of 0.3 µg/g (DW), except for calendar week 21 (May, 0.03 ± 0.01 µg/g (DW)). Concentrations analyzed with plant fresh weight followed the same trend but were only 20–49% of the concentrations determined in plant dry weight (averaged approx. 30% of dry weight).

Lolitrem B concentrations exceeded the toxicity threshold of 1.8 µg/g (DW) only in calendar week 29 (July, 2.00 ± 0.32 µg/g (DW) and 33 (August, 6.57 ± 1.09 µg/g (DW)) (Figure 2, Appendix A). Concentration changed with season and peaked in calendar week 33 (in August) for both fresh and dry weight. (Figure 2). Concentrations analyzed with plant fresh weight were only 13–40% of the concentrations determined in plant dry weight (averaged approx. 24% of dry weight).

Paxilline concentration decreased after plant transfer to the common garden to concentrations bellow the detection limit but increased from calendar week 31 until calendar week 37 (September) up to 0.56 ± 0.28 µg/g (DW) (Figure 2, Appendix A). Concentrations analyzed with fresh weight were below detection limit.

Detected differences between alkaloid concentrations analyzed from plant fresh and dry weight were higher in summer than in spring or autumn (Figure 2).

### 3.4. Comparison of Two Alkaloid Detection Methods

Concentrations (DW) based on the method from [36] (method 2) were significantly higher for peramine (averaged approx. 2.2 times) compared to the method described in this study (Method 1) (paired *t*-test (*n* = 20): t = −3.34, df = 9, *p* = 0.008 **) (Figure 3). In contrast, ergovaline concentrations (DW) were significantly lower (averaged approx. 0.42 times) when detected with method 2 compared to method 1 (paired *t*-test (*n* = 19): t = −2.82, df = 8, *p* = 0.02 *) (Figure 3). Lolitrem B concentrations (DW) were marginally higher (averaged approx. 1.7 times) when detected with method 2 compared to method 1 (paired *t*-test (*n* = 19): t = 2.23, df = 9, *p* = 0.052) (Figure 3).

Comparing each sample with the two methods showed that some samples showed an opposite trend (peramine: sample *d*, ergovaline: sample *j*, lolitrem B: sample *b*, *d*) (Figure 3). Peramine, ergovaline, and lolitrem B were found in each sample when analyzed with method 1. There was no detection of ergovaline in sample *f* and no detection of lolitrem B in sample *b* when using method 2.

## 4. Discussion

### 4.1. Field Study in Germany

Our results are in accordance with the results found in König et al. (2018) that land-use intensity, mowing, grazing, or fertilization had no influence on infection rates and alkaloid concentrations of *L. perenne* infected with *E. festucae* var. *lolii* in three regions in Germany, even though our results are now based on plant dry weight (i). Although other studies showed influence of grazing [28,29,52,53] or fertilization [46,52] on alkaloid concentrations, we detected no significant effects in our field study sites. However, in most of the studies, results are based on functional treatments from common garden experiments, whereas our study represents results from real world agricultural fields, highlighting the importance of authentic field studies. Infection rates differed significantly between the three regions in this study and in König et al. (2018), but this could not be explained by land-use intensity, mowing, grazing, or fertilization. König et al. (2018) also showed that sowing had no influence on infection rates or alkaloid concentrations. However, there are hints that commercially available seed mixtures can contain *Epichloë*-infected seeds in Germany [31], which might lead to increasing *Epichloë* infections. In contrast to König et al. (2018), we extracted alkaloids from plant dry weight with an altered quantitation method [31]. We showed that intoxication risk on population level on German grasslands is similarly low using plant dry weight compared to fresh weight [23]. Single samples exceeded toxicity thresholds in both studies, but not on population level. We assume that the difference in alkaloid concentrations often dilutes on population level in Germany similar to other regions, e.g., Spain [54].

### 4.2. Alkaloid Concentrations throughout the Year in a Field Study

We monitored alkaloid concentrations in *Lolium perenne* infected with *Epichloë festucae* var. *lolii* over ten months and showed an increase in alkaloid concentrations detected in dried plant material in summer (ii). Other studies also indicated that alkaloid concentrations peak in late summer [23,55,56,57]. Our results represent the course of alkaloid concentrations in a real world field without additional watering, compared to other studies, where observations of alkaloid concentrations over several months are often highly controlled, e.g., performed on grasslands sown with *Epichloë* endophyte infected seeds [56], grasses from the field, but selected for *Epichloë* infections, and potted [57], or potted grasses grown from *Epichloë* infected seeds [55].

On field sites, we only detected lolitrem B and peramine, but not ergovaline, which is consistent with the results of a previous study, showing that the starting gene for the ergovaline biosynthesis pathway *dmaW* is missing in *E. festucae* var. *lolii* in *L. perenne* in the same German grasslands [24]. Another study showed that ergovaline concentrations in wild *Festuca rubra* infected with *Epichloë festucae* varied depending on geographic origin, and chemical profiles changed when plants were translocated, which might also be a reason for the absence of ergovaline [58]. We also did not detect paxilline, a precursor of lolitrem B. That indicates that the amount of paxilline was below the detection limit, and the precursor was probably immediately converted to lolitrem B. Paxilline is currently mostly neglected in studies concerning endophyte-mediated toxicity. Due to its tremorgenic effects on vertebrates, paxilline should be considered in future intoxication studies [9,24]. We calculated alkaloid concentrations on the population level (mean of infected and not infected *L. perenne* samples per study site) and of only infected samples (mean of infected individual plants per study site), which would reflect alkaloid concentrations in a monoculture with only infected *L. perenne* plants. Concentrations of insect-deterring peramine on the population level exceeded the toxicity threshold of 2 µg/g (DW) [12] only in September. However, mean alkaloid concentrations per study sites for infected plants showed exceeding concentrations throughout the entire summer, which can be a benefit for farmers due to herbivore protection. Considering the deterring threshold of 15 µg/g (DW) as postulated for the deterrence of Argentine stem weevil [14,15], none of the tested samples exceeded this threshold.

Lolitrem B concentrations exceeding toxicity thresholds of 1.8 µg/g (DW) can lead to intoxication events in grazing livestock by causing ryegrass staggers [16]. On the population level, lolitrem B never exceeded the toxicity threshold, whereas it did in September, when only considering infected samples for the calculation of mean alkaloid concentrations per study site. Therefore, intoxication of livestock due to lolitrem B might be possible in late summer on *L. perenne* monocultures, with a high frequency of *Epichloë* infection.

Seasonal variation is also known from carbohydrates, like fructane and simple sugars in orchardgrass (*Dactylis glomerata*), which both peak in spring [59]. High levels of fructane can cause laminitis in horses [59].

*Epichloë* infection rates seem to differ also with the season but are in fact caused by an increasing amount of the fungus in the host grass in summer [55], which increases the detectability of the infection. Therefore, immunoblot assays should not be used alone for field screenings, because the intensity of the color reaction of the immunoblot assay decreases with lower amount of fungus and misidentifications with other fungal infections are possible [60]. In our study, we additionally used HPLC analyses to confirm an infection. Alkaloid concentrations correlate with the amount of the fungus in the plant, but other factors such as the nature of the plant tissue and specific regulations for each alkaloid might also strongly affect alkaloid concentrations [47].

### 4.3. Common Garden Study

We conducted a common garden study, monitoring alkaloid concentrations over a ten month period following the setup of [55] to compare alkaloid concentrations between samples extracted from fresh and dried plant material. In this experiment, we used *L. perenne* seeds infected with *E. festucae* var. *lolii* from New Zealand, which are known to produce vertebrate toxic lolitrem B and ergovaline, as well as insect-deterring peramine. We showed that alkaloid concentrations increased in summer and decreased in winter (iii) independent of plant dry or fresh weight (iv). However, alkaloid concentrations analyzed with fresh weight were lower compared to dry weight, which was previously shown for paxilline and ergot alkaloids [61]. In the study of Bauer et al. (2018), the authors showed that alkaloid concentrations for paxilline and ergot alkaloids were approximately ten times higher when analyzed with dry weight instead of fresh weight. The dried plant weight was also around 10% of the plant fresh weight [61]. In our study, the loss on drying also corresponded to the difference in alkaloid concentrations based on fresh or dry weight. That shows that the difference in alkaloid concentrations can be explained by the loss of water during drying. The study of Bauer et al. (2018) used commercially available grass seeds from German horticulture shops, hence *Epichloë* infection status of the seeds were unknown before [61]. These results show that using fresh weight concentrations lead to an underestimation of toxic alkaloid concentrations. Most studies use dry weight to measure alkaloid concentrations, with some exceptions due to specific constraints, e.g., because of limited sample amount [23] or parallel measurements, which require fresh plant material [55]. Nevertheless, alkaloid concentrations in dried plant material are more accurate because of possible varying water contents in the plants.

Alkaloid concentrations in dried plant material decreased after transfer of the plants from the greenhouse to the common garden before alkaloid concentrations increased again, which might be explained by an adaptation period to common garden conditions. Another possible explanation might be that fungal growth is delayed due to fast plant growth, which is reflected in a temporary decrease in alkaloid concentrations [14,62]. Peak alkaloid concentrations in dried and fresh plant material found during late summer (September) correlate with peak abundances of main grass herbivores, such as species from the insect order Orthoptera [63], which could explain the peak in insect-deterring peramine concentration in late summer [53]. Ryegrass stagger outbreaks occurred in summer and autumn in Australia and New Zealand, which can be explained by peaking lolitrem B concentrations [8]. In the study of [55], ergovaline and lolitrem B concentrations based on plant fresh weight exceeded toxicity thresholds only in the third year. Here, we showed that ergovaline and lolitrem B concentrations in dried plant material exceed toxicity thresholds already in the first year. In another study, it was shown that variation in lolitrem B concentrations was low over three years, whereas ergovaline concentrations varied and might be influenced by abiotic factors [57]. Our study allows a rough estimate to compare fresh and plant dry weight for alkaloid concentrations in *L. perenne* with a correction multiplication of 3.12 for peramine concentrations, 3.55 for lolitrem B, and 3.65 for ergovaline (fresh weight to dry weight). However, care must be taken, e.g., due to a climate change-mediated increase in extreme events, such as heavy rains and persistent droughts with tremendous fluctuations in plant water content [64].

### 4.4. Comparison of Two Alkaloid Detection Methods

There are different methods to detect and quantify alkaloids produced by *Epichloë* species in cool season grass species [16,20,32,34,36]. Here, we compared two UPLC methods differing in the length of their used column and the additional detection of paxilline in method 1. In method 1, we used to detect alkaloid concentrations in this publication, whereas method 2 was used in previous publications [23,36,53,55,62]. Alkaloid concentrations detected with method 2 were higher for peramine and lolitrem B, whereas alkaloid concentrations detected with method 1 were higher for ergovaline (v). The limit of detection is comparable between the methods and cannot explain the differences. Both methods used the same solvents and column materials, hence the efficiency of ionization should be comparable, although the longer column in method 1 and the differing gradient changes the solvent composition slightly, which might have led to the differences in alkaloid detection. Homoperamine was used as a reference to quantify lolitrem B in both methods [31,36], although it is not structurally related to lolitrem B. Lolitrem B was only available in small amounts for method development, but not for a validation of the method. The substance ergovaline used for validation was only available for method 1. Hence, we assume that detection of ergovaline is more reliable with this method. Otherwise, method 2 detected higher lolitrem B and peramine concentrations. We developed method 1 originally, because we wanted to detect lolines, lolitrem B, ergovaline, peramine, and paxilline within one method. Lolines belong to the 1-aminopyrrolizidines, another group of alkaloids, which can be produced by some *Epichloë* endophytes such as *E. uncinata* in *Festuca pratensis* and which have insect toxic properties [65]. We used a previously published method [32] to add the detection of polar lolines and lipophilic paxilline, which extends the run time of the method. We had to adapt the method from [32] due to a low solubility of lolitrem B and paxilline in aqueous solutions and a better ionization of alkaloids with formic acid instead of ammonium formate. The detection of molecules with differing molecule size and polarity can lead to methodological differences like solubility or ionization problems, which could result in differing alkaloid concentrations. Differences in chromatographic conditions (different solvent gradient and column dimensions) can lead to different ionization efficiencies and other matrix effects, resulting in different alkaloid concentrations. In order to determine absolute alkaloid concentrations, it would be necessary to use isotope-labelled reference substances, which are not available for the tested alkaloids. Due to that, toxicity thresholds from literature are also not determined with isotope-labelled reference substances (if mass spectrometry is used as detection method), which means that only a rough comparability of detected alkaloid concentrations with toxicity thresholds is possible. In a previous study, we compared alkaloid concentrations in seed mixtures with two different methods [31]. Alkaloid concentrations were detected with method 1 from this study and another UPLC method of the Endophyte Service of Oregon State University. Absolute numbers of alkaloid concentrations differed in this study too, but detection of alkaloids (presence/absence) was consistent for all samples [31]. It can be assumed that other methods of alkaloid detection differ in their absolute amounts of alkaloid concentrations and can only be compared relatively. Bauer et al. (2018), for example, showed that paxilline concentrations detected with HPLC-MS/MS were less than 3% of the paxilline level detected with enzyme immunoassay [35], but this might be explained due to the often unselective detection of immunoassays [60]. Hence, only validated methods should be used for the detection of alkaloids to compare alkaloid concentrations with other studies and toxicity thresholds from literature. This can be even more important with increasing temperatures due to climate change, which can increase alkaloid production by *Epichloë* spp. [6,66,67].

## 5. Conclusions

We showed that (i) land-use intensity had no effect on alkaloid dry weight concentrations. We also showed that (ii) the German grasslands contained grass individuals with concentrations above the toxicity level, but not on a population level of all *L. perenne* plants per study sites. Both results confirmed the findings of a previous study [23]. Previous findings on seasonal trends in alkaloid concentration using plant fresh weight [55] could be (iii) confirmed with plant dry weight experiments. We showed that (iv) alkaloid dry weight concentrations in *L. perenne* plants are approximately 3–4 times higher compared to fresh weight concentrations. Finally, we showed that (v) detection methods differed in alkaloid concentrations, due to the use of different chromatographic conditions. We assume that the influence of the alkaloid detection method is low for general trends in a functional study, whereas to estimate intoxication risk for livestock, a validated method is important.

## Figures and Tables

**Figure 1 jof-06-00177-f001:**
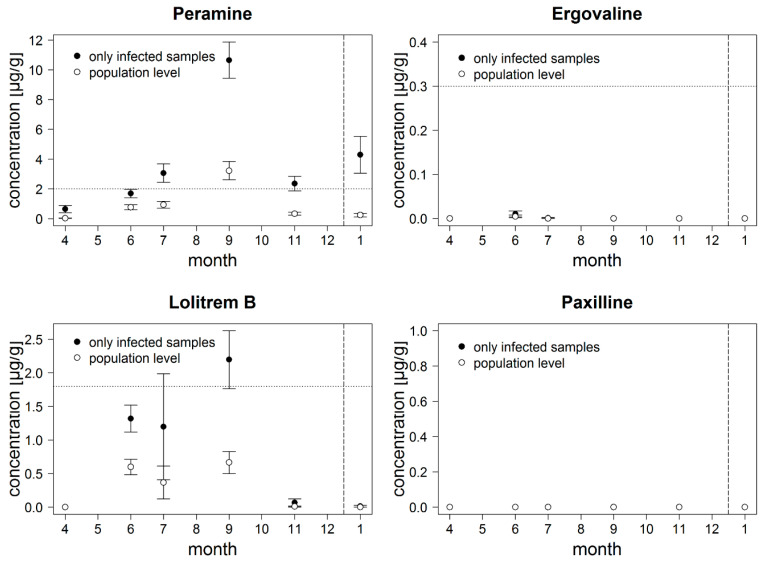
Concentrations of peramine and lolitrem B in one year in 2018 and the beginning of 2019 on five study sites in Schorfheide-Chorin, Germany, on population level (unfilled dots: µg/g dry weight (DW), mean ± SE, *n* = 575). Ergovaline and paxilline were not detected in the samples. Filled dots show only infected samples (mean ± SE, *n* = 133). Dotted horizontal line indicates toxicity threshold of the alkaloid based on dry weight and dashed vertical line indicates the turn of the year.

**Figure 2 jof-06-00177-f002:**
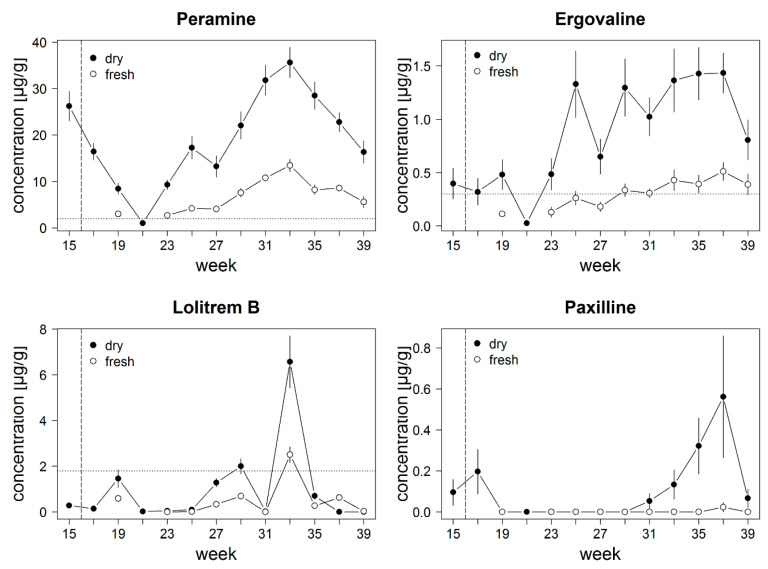
Alkaloid concentrations in [µg/g] (mean ± SE) in *L. perenne* plants infected with *Epichloë festucae* var. *lolii* during several weeks (calendar weeks) in a common garden experiment analyzed with dry (filled circles) and fresh (empty circles) plant weight. Plants were sampled every two weeks from the beginning of April until the end of September. The first sampling (calendar week 15) was performed in the greenhouse, afterwards the plants were placed outside in a common garden (marked with a vertical dashed line). Toxicity thresholds based on dry weight are marked with a horizontal dotted line.

**Figure 3 jof-06-00177-f003:**
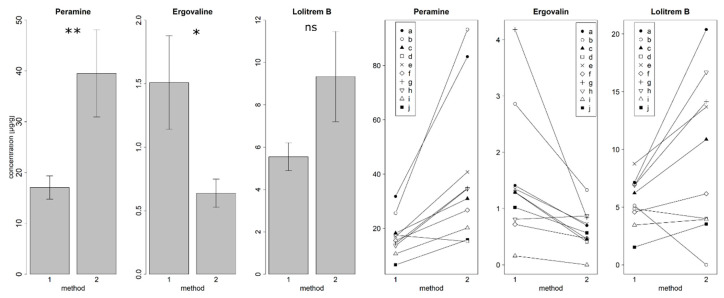
a: Comparison of alkaloid concentrations [µg/g DW] (peramine, ergovaline, and lolitrem B, mean ± SE) detected with the UPLC method from this study (method 1) and the UPLC method from [36] (method 2). Peramine: *n* = 10, *p* = 0.008 **, ergovaline: *n* = 9, *p* = 0.02 *, and lolitrem B: *n* = 10, *p* = 0.052, not significant). b: Comparison of alkaloid concentrations [µg/g] (peramine, ergovaline, and lolitrem B) of single samples (*a*–*j*) detected with the UPLC method from this study (method 1) and the UPLC method from [36] (method 2).

**Table 1 jof-06-00177-t001:** Generalized linear models (infection rates) and linear mixed effect models (peramine and lolitrem B). For the alkaloid concentrations (peramine and lolitrem B), only infected samples were used for analyses. LUI: land-use intensity; Mowing/Grazing/Fertilization (y/n): Mowing/Grazing/Fertilization: yes/no; Mowing tot: frequency of mowing per year; Grazing tot: the grazing intensity reflected by the density of livestock (livestock units days of grazing ha^−1^ year^−1^); Fertilization tot: fertilization level (kg nitrogen ha^−1^ year^−1^). *** = highly significant.

	Infection Rates			Peramine			Lolitrem B		
	χ^2^ dF	χ^2^	*p*	dF	F	*p*	dF	F	*p*
**Region**	**2**	**17.45**	**<0.001 *****	2, 107	1.84	0.19	2, 75	2.77	0.11
**LUI**	1	0.53	0.47	1, 107	3.21	0.10	1, 75	0.24	0.64
**Mowing (y/n)**	1	0.09	0.76	1, 107	0.004	0.95	1, 75	0.01	0.94
**Mowing tot**	1	0.09	0.76	1, 107	0.10	0.75	1, 75	0.001	0.98
**Grazing (y/n)**	1	0.81	0.37	1, 107	0.08	0.79	1, 75	0.02	0.90
**Grazing tot**	1	0.12	0.73	1, 107	0.22	0.64	1, 75	0.01	0.91
**Fertilization (y/n)**	1	1.70	0.19	1, 107	0.33	0.58	1, 75	0.15	0.70
**Fertilization tot**	1	0.36	0.55	1, 107	4.14	0.08	1, 75	0.09	0.78

**Table 2 jof-06-00177-t002:** Number of individuals (IND, only infected *Lolium perenne*) or study sites (SITE, infected and not infected *Lolium perenne*) within alkaloid concentration range [µg/g DW] of peramine, ergovaline, lolitrem B, and paxilline and their relative proportion of all individuals, or all study sites. Max: maximum of alkaloid concentrations. Individuals or study sites that show concentrations above toxicity thresholds are marked in bold. Individuals: *n* = 131, study sites: *n* = 66.

	Individual Concentrations	Mean Concentrations per Study Site
Peramine	Ergovaline	Lolitrem B	Paxilline	Peramine	Ergovaline	Lolitrem B	Paxilline
Conc. (µg/g)	IND	%	IND	%	IND	%	IND	%	SITE	%	SITE	%	SITE	%	SITE	%
**0.0**	4	3.1	130	99.2	54	41.2	130	99.2	38	57.6	65	98.5	42	63.6	65	98.5
**0–0.3**	13	9.9	1	0.8	24	18.30	1	0.8	15	22.7	1	1.5	23	34.9	1	1.5
**0.3–1**	52	39.7	**0**	**0.0**	32	24.4	0	0.0	9	13.6	**0**	**0.0**	1	1.5	0	0.0
**1.0–2.0**	36	27.5	**0**	**0.0**	10	7.6	0	0.0	4	6.1	**0**	**0.0**	0	0.0	0	0.0
**2.0–3.0**	**10**	**7.6**	**0**	**0.0**	**4**	**3.1**	0	0.0	**0**	**0.0**	**0**	**0.0**	**0**	**0.0**	0	0.0
**3.0–5.0**	**10**	**7.6**	**0**	**0.0**	**3**	**2.3**	0	0.0	**0**	**0.0**	**0**	**0.0**	**0**	**0.0**	0	0.0
**5.0–10.0**	**4**	**3.1**	**0**	**0.0**	**0**	**0.0**	0	0.0	**0**	**0.0**	**0**	**0.0**	**0**	**0.0**	0	0.0
**>10.0**	**2**	**1.5**	**0**	**0.0**	**0**	**0.0**	0	0.0	**0**	**0.0**	**0**	**0.0**	**0**	**0.0**	0	0.0
**Max (µg/g)**	12.0		0.17		4.2		0.15		1.7		0.0		0.5		0.01

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
