# Peer review of "Alkaloid Concentrations of *Lolium perenne* Infected with *Epichloë festucae* var. *lolii* with Different Detection Methods—A Re-Evaluation of Intoxication Risk in Germany?"

_jof, 2020, doi:10.3390/jof6030177_

Round 1

Reviewer 1 Report

The work entitled “Alkaloid concentrations of Lolium perenne infected with Epichloë festucae var. lolii with different detection methods – a re-evaluation of intoxication risk in Germany?” tries to cover an important flaw in field studies, their standardization, as well as their applicability locally. At a time of greater need to assess the conditions and practices of agriculture and its impact on the environment and health of cattle and humans. Moreover Vikuk and co-authors address an important issue, the balance between the use of natural compounds naturally present in the culture versus their toxicity. For all that justifies the importance of the present work.

Although in general the work does not present major flaws, there are some points that could be addressed and that would enrich the value and relevance of the work.

- the role of these alkaloids in the human food chain. The accumulation of these compounds in cattle and consecutively their bioavailability for humans (e.g. Tingting Wang et al, https://doi.org/10.1016/j.foodcont.2018.11.033)

- in these studies, exposure to levels of infection is important. The graph in supp data should appear in the main text.

- relationship between infection levels / seasonal variation of compounds.

Positive the seasonal variation of compounds Peramine and Lolitrem B presented, however the question could be better explored, even uninfected plants show this variation.

Author Response

Although in general the work does not present major flaws, there are some points that could be addressed and that would enrich the value and relevance of the work.

- the role of these alkaloids in the human food chain. The accumulation of these compounds in cattle and consecutively their bioavailability for humans (e.g. Tingting Wang et al, https://doi.org/10.1016/j.foodcont.2018.11.033)

Reply: Lolitrem B and ergovaline are vertebrate toxic alkaloids, thus they can be also toxic for humans. Since these alkaloids are only produced by specific Epichloë fungi infecting cool season grass species (e.g. Epichloë festucae var. lolii infecting Lolium perenne which we investigated in this study), the toxicity is mainly relevant for grazing animals, like cattle, sheep or horses. Grasses are pollinated by wind, hence contamination of honey, like with pyrrolizidine alkaloids in the study that you added, is very unlikely. Nevertheless, the question of the bioavailability for humans is justified, since humans consume animal products, like meat and milk from cows or sheep, which can be toxified by these alkaloids in the grasses. The intoxication risk of humans due to consumption of animal products such as milk or meat are low, as concentrations of alkaloids are strongly below the toxicity thresholds (Finch et al., 2012, 2013). We added this information now in the introduction (lines 56-60):

“Toxic secondary plant compounds can be also a thread in the human food chain, like pyrrolyzidinalkaloids in honey (Wang et al., 2019). But studies showed, that concentrations of Epichloë-alkaloids in the milk of intoxicated cows or in sheep fat are below the toxicity threshold, indicating that the intoxication risk of humans consuming animal products such as milk or meat is low (Finch et al., 2012, 2013).”

- in these studies, exposure to levels of infection is important. The graph in supp data should appear in the main text.

Reply: Seasonal changes in the level of infection is in fact a methodical artefact. Grass individuals can be either infected or not with the asexual Epichloë endophyte and transmission occurs exclusively via seeds. Infection rates of populations can therefore increase or decrease only from one generation to the next generation, but usually not within the lifespan of one individual. We added one paragraph in the discussion to explain why infection rates seem to change with season (lines 426-433): “Epichloë infection rates seem to differ also with season, but is in fact caused by an increasing amount of the fungus in the host grass in summer (Fuchs et al., 2017), which increases the detectability of the infection. Therefore, immunoblot assays should not be used alone for field screenings, because the intensity of the colour reaction of the immunoblot assay decreases with lower amount of fungus and misidentifications with other fungal infections are possible (Jensen et al., 2011). In our study, we additionally used HPLC analyses to confirm an infection. Alkaloid concentrations correlate with the amount of the fungus in the plant, but other factors like the nature of the plant tissue and specific regulations for each alkaloid might also strongly affect alkaloid concentrations (Spiering et al., 2005).”

We deleted table S3 from the supplementary, because it describes this methodical artefact and could confuse readers.

- relationship between infection levels / seasonal variation of compounds.

Reply: While compounds and the abundance of the fungus vary with season, there is no seasonal variation of infection levels of individuals. It is a methodical artefact, which we explained above.

Positive the seasonal variation of compounds Peramine and Lolitrem B presented, however the question could be better explored, even uninfected plants show this variation.

Reply: Peramine and Lolitrem B were detected in Lolium perenne plants infected with Epichloë festucae var. lolii and can only be produced by this symbiosis and not in Epichloë free plants. However, other grass compounds show similar seasonal variations. For example, carbohydrates, like fructane and simple sugars in orchardgrass (Dactylis glomerata) which both peak in spring (Kagan et al., 2011). High levels of fructane can cause  laminitis in horses (Kagan et al., 2011). We added this information in the discussion (lines 423-425).

Literature:

Finch, S.C., Fletcher, L.R., and Babu, J.V. (2012). The evaluation of endophyte toxin residues in sheep fat. New Zealand Veterinary Journal 60, 56–60.

Finch, S.C., Thom, E.R., Babu, J.V., Hawkes, A.D., and Waugh, C.D. (2013). The evaluation of fungal endophyte toxin residues in milk. New Zealand Veterinary Journal 61, 11–17.

Fuchs, B., Krischke, M., Mueller, M.J., and Krauss, J. (2017). Plant age and seasonal timing determine endophyte growth and alkaloid biosynthesis. Fungal Ecology 29, 52–58.

Jensen, J.B., Gonzalez, V.T., Guevara, D.U., Bhuvaneswari, T.V., Wali, P.R., Tejesvi, M.V., Pirttila, A.M., Bazely, D., Vicari, M., and Brathen, K.A. (2011). Kit for detection of fungal endophytes of grasses yields inconsistent results. Methods in Ecology and Evolution 2, 197–201.

Kagan, I.A., Kirch, B.H., Thatcher, C.D., Strickland, J.R., Teutsch, C.D., Elvinger, F., and Pleasant, R.S. (2011). Seasonal and Diurnal Variation in Simple Sugar and Fructan Composition of Orchardgrass Pasture and Hay in the Piedmont Region of the United States. Journal of Equine Veterinary Science 31, 488–497.

Rudolph, W., Remane, D., Wissenbach, D.K., and Peters, F.T. (2018). Development and validation of an ultrahigh performance liquid chromatography-high resolution tandem mass spectrometry assay for nine toxic alkaloids from endophyte-infected pasture grasses in horse serum. Journal of Chromatography A 1560, 35–44.

Spiering, M.J., Lane, G.A., Christensen, M.J., and Schmid, J. (2005). Distribution of the fungal endophyte Neotyphodium lolii is not a major determinant of the distribution of fungal alkaloids in Lolium perenne plants. Phytochemistry 66, 195–202.

Wang, T., Frandsen, H.L., Christiansson, N.R., Rosendal, S.E., Pedersen, M., and Smedsgaard, J. (2019). Pyrrolizidine alkaloids in honey: Quantification with and without standards. Food Control 98, 227–237.

Reviewer 2 Report

Dear Authors,

The time of chromatographic analysis is very long (25 min) for the UPLC-MS / MS system used. What was the reason for using such a long time to analyze a single sample when analyzing only a few analytes?

I would suggest to insert the chromatograms for the analyzed compounds obtained for both tested methods.

Writing that several samples exceeded the toxicity threshold it might be better to present the results for these samples in tabular form, I think it would be more readable to the reader.

I am glad that the authors compared the two chromatographic methods, but with a system such as the UPLC-MS / MS, it would be wiser to optimize the chromatographic method by using a matrix calibration curve for all compounds presented in the study in one analytical protocol (this is only my suggestion).

Reviewer 3 Report

This is a well-written paper reporting on alkaloid content in cool season grasses in Germany.

It compares methods and presents somewhat conflicting results for method 1 and 2. The explanation for the conflicting results was insufficient. Please do an additional experiment to better answer if the results are due to column length or some other reason.

One minor point - please use the term plant dry weight instead of dry plant weight.

Author Response

This is a well-written paper reporting on alkaloid content in cool season grasses in Germany.

We thank the reviewer for the comments on our manuscript.

It compares methods and presents somewhat conflicting results for method 1 and 2. The explanation for the conflicting results was insufficient. Please do an additional experiment to better answer if the results are due to column length or some other reason.

Reply: Both chromatographic separation techniques differ only in the column length and the composition of the mobile phase during the analysis, whereas mobile phase A and B were identical as well as the ionization parameters for all analytes and standards. However, mass spectrometry is highly sensitive to matrix effects, which could directly affect the ionization efficiency of all compounds, resulting in slightly different signal intensities for some of these compounds, depending on the different solvent gradients used in both respective methods. We agree that method 1 and 2 show somehow conflicting results. And more experiments would be necessary to find a causality. But we have to leave it to speculations, because resubmission time is too short to conduct additional experiments. We think that we can still show that there might be variations between validated methods, which we want to call attention to.

One minor point - please use the term plant dry weight instead of dry plant weight.

Reply: We changed the term dry plant weight and fresh plant weight in plant dry weight and plant fresh weight throughout the manuscript.

Round 2

Reviewer 3 Report

Minor editing is needed for grammar for the new sections.